# Constrained bit allocation for neural networks

Souleyman Boudouh
EPFL, MBZUAI
Laussanne, Switzerland
souleyman.boudouh@epfl.ch

Simla Burcu Harma
EPFL
Laussanne, Switzerland
simla.harma@epfl.ch

Abdulrahman Mahmoud
MBZUAI
Abu Dhabi, United Arab Emirates
abdulrahman.mahmoud@mbzuai.ac.ae

Babak Falsafi
EPFL
Lausanne, Switzerland
babak.falsafi@epfl.ch

## Abstract

The increasing complexity of deep neural networks (DNNs) necessitates effective model compression to reduce their computational and memory footprints for deployment on resource-constrained hardware. Layer-wise bit allocation is a prominent compression method shown to significantly reduce DNN footprints while preserving model accuracy. However, how best to incorporate hardware constraints within the allocation search remains a key question, as many tacitly assume constraints can be adequately handled via soft penalties or heuristics, often failing to guarantee feasibility or optimality. In this paper, we explore a reformulation of the bit allocation problem as an explicit constrained optimization problem, solved using interior-point methods within a NAS-based framework, notably requiring only minimal calibration data (as few as 128 samples). We corroborate this approach with experiments spanning transformer architectures (Llama, Gemma, Qwen; 500M-3B parameters), evaluating performance with MXFP formats. We show that this constrained formulation not only allows us to achieve significantly finer resolution in compression ratios compared to the discrete steps offered by uniform MXFP application (e.g., 4.25, 6.25, 8.25 bits), but also demonstrates that explicitly satisfying hardware budgets while optimizing for accuracy consistently outperforms uniform allocation methods, improving performance by up to several standard deviations in some cases, especially under strict resource limits. Our findings extend to the efficient deployment of large models in resource-constrained compute platforms, offering insights into best practices for applying bit allocation to maximize hardware resource efficiency without unduly compromising accuracy.

## Keywords

Neural networks, Compression, Numerical formats, Block floating point, Microexponents, MXFP, Interior point methods, Constrained optimization, Bit allocation, Mixed precision

## 1 Introduction

Recent breakthroughs in deep neural networks (DNNs) have achieved unprecedented performance across tasks like computer vision and natural language processing. However, this progress coincides with exponential increases in model size and computational demands, with state-of-the-art models requiring vast memory and compute resources, challenging deployment on edge devices and constrained

'ISCA 2025', June 21–25, 2025, Tokyo, Japan
2024. ACM ISBN 978-1-4503-XXXX-X/18/06
https://doi.org/XXXXXXX.XXXXXXX

accelerators. Significant efforts were made to develop effective model compression techniques, such as lower precision data formats, that preserve accuracy while reducing resource requirements.

To support compressed inference and training, researchers and hardware consortia have proposed various reduced-precision numerical formats [7, 9, 23]. These formats range from traditional integers (INT8/4) [13] and custom floating-point types (FP8/6) [18] to block-level schemes like BFP [9], MXFP, and MXINT [7, 23]. These low-precision data formats significantly improve arithmetic density and energy efficiency, often by amortizing exponent costs. Their adoption in accelerators like NVIDIA's Hopper and Google's TPUs [7, 16–18] highlights their critical role in efficient DNN deployment. Despite advances in format design, DNN layers exhibit varying sensitivity to reduced precision [14, 32], making uniform format application often suboptimal for accuracy-efficiency tradeoffs. The disparity in compression sensitivity across layers motivated differentiable neural architecture search (DNAS) methods [3, 4, 29, 30] that automate layer-wise precision assignment by optimizing soft architectural parameters jointly with weights. However, these DNAS approaches typically operate in a training-aware setting, requiring full access to training data and tuning of the model weights.

Although powerful, existing DNAS frameworks often handle hardware constraints via soft penalties or heuristics, lacking principled guarantees on the final complexity-performance tradeoff and often necessitating extensive Pareto front exploration. While the use of barrier functions has been previously proposed for quantization-aware training methods [31], we propose to revisit this challenge through the lens of constrained optimization for post training quantization, reformulating the problem to treat hardware requirements as explicit constraints. Our approach integrates model complexity directly using a regularizer based on the interior-point method [2, 19] to systematically drive solutions towards feasibility.

Our framework targets the post-training setting, using frozen weights and only a small calibration dataset. It operates on relaxed softmax-distributed architectural parameters and employs an annealed regularization schedule to efficiently solve the constrained optimization problem. This yields format allocations satisfying resource budgets exactly while optimizing performance. Notably, our approach offers finer resolution in compression ratios and predictable model behavior across resource settings without retraining or finetuning. Our contributions are as follows:

- We revisit DNAS approaches for mixed-precision schemes and introduce a novel interior-points based optimization

framework for differentiable post-training architecture search that enables hardware-constrained bit allocation without requiring retraining or fine-tuning of the original model weights.

- We formulate a regularized objective that supports smooth, feasible layer-wise precision allocation using only a small calibration dataset, making the method suitable for deployment in low-data settings.
- We demonstrate that smart allocation of precision across layers leads to significant improvements in few-shot evaluation tasks and maintains competitive performance in zero-shot settings, even under strict hardware budgets.
- Our method provides fine-grained control and higher resolution over intermediate compression rates with predictable and stable performance trends.

## 2 Related work

***Low precision data formats***. Recent advancements in low-precision data representations have made it possible to train and deploy deep neural networks (DNNs) using low precision formats, drastically reducing compute cost and memory footprint. A prominent direction in this area is block-based numerical formats, which group multiple elements under a shared exponent, thus improving arithmetic density while maintaining sufficient dynamic range. Drumond et al. proposed Hybrid Block Floating Point (HBFP) [9], a representation that performs all dot-product operations using block floating point arithmetic while retaining standard floating point for element-wise functions and control logic. This hybrid strategy ensures convergence comparable to full-precision (FP32) training across a variety of workloads. They demonstrate that HBFP can match the accuracy of FP32 while achieving up to 8.5× throughput gains over FP16, offering a drop-in replacement suitable for deployment with modest hardware modifications.

More recently, Darvish Rouhani et al. developed the MX (Microscaling) data format family [7, 23], designed to support both inference and training by combining narrow data types (e.g., INT8, FP6, FP4) with fine-grained block-level scaling. These formats are tailored for high-performance computing environments and provide a tunable balance between computational efficiency, numerical stability, and usability. MX formats are evaluated across a wide range of discriminative and generative tasks and are shown to preserve model fidelity even when applied to large-scale transformers with sub-8-bit activations, weights, and gradients. They also demonstrate compatibility with common training pipelines and require minimal adjustment to hyperparameters or infrastructure. As a result, MX formats are gaining support in next-generation hardware [6, 20]. These advances build upon earlier work that established block-based representations as a viable path to high-efficiency training, with design techniques such as tiling and wide weight storage shown to mitigate the precision loss risks typically associated with narrow mantissas.

***Differentiable Neural Architecture Search***. The optimal bit allocation problem for neural network is the following: Let a generic neural network of $L$ layers labeled $l \in \mathbb{L} = \{1, ..., L\}$ with associated weights $W = \{W^l\}_{l=1}^L$. Assuming we have access to a set of

compression schemes labelled $Q_d(.)$ with $d \in \mathbb{D} = \{1, ..., D\}$, we get:

$$
\begin{aligned}
\hat{W}^l &\stackrel{def}{=} \sum_d^D A_d^l Q_d(W^l) \\
s.t. \quad &\sum_{d=1}^D A_d^l = 1 \quad \forall l \in \mathbb{L} \\
&A_d^l \in \{0, 1\} \quad \forall (l, d) \in \mathbb{L} \times \mathbb{D}
\end{aligned}
\tag{1}
$$

Equation (1) represents the core formulation used in differentiable mixed-precision search frameworks [3–5, 29, 30]. We want to find the optimal decision variables $A_d^l$ solving the multi-objective minimization problem:

$$
\begin{aligned}
\min_{\hat{W}, A} \quad &\mathcal{L}(A, \hat{W}) \\
\min_{A} \quad &C(A, \hat{W})
\end{aligned}
\tag{2}
$$

Here $\mathcal{L}(A, \hat{W})$ is the model loss function, and $C(A, \hat{W})$ is defined as a complexity cost on the architecture $A$, often related to hardware constraints such as size or latency. Throughout this paper, we set $C(A, \hat{W})$ to be the average bit-width per element of the target model. The exponential number of possibilities for a choice of $A$ and the latency induced by evaluating $\mathcal{L}$ make it difficult to efficiently solve the problem using combinatorial techniques. Taking inspiration from approximation algorithms, a popular approach is to relax the conditions on $A$ and interpet $A^l$ as a probability distribution.

$$
\begin{cases} \sum_{d=1}^D A_d^l = 1 \\ A_d^l \in \{0, 1\} \end{cases} \implies \begin{cases} \sum_{d=1}^D A_d^l = 1 & \forall l \in \mathbb{L} \\ A_d^l \geq 0 & \forall (l, d) \in \mathbb{L} \times \mathbb{D} \end{cases}
\tag{3}
$$

To that end, Neural Architecture Search frameworks introduced the following paramterization of $A^l$ in terms of logits $\{x^l\}_{l=1}^L \subset \mathbb{R}^D$:

$$
A_d^l \equiv A_d^l(x^l) = \frac{exp(x_d^l)}{\sum_{k=1}^D exp(x_k^l)}
\tag{4}
$$

NAS-inspired frameworks for mixed precision quantization leverage the above relaxation by building a super-network, for which they train the weights and architectural parameters alternatively, handling the search for two sets of parameters at once. Once done, a feasible solution $\bar{A}$ to the original decision problem is rounded from the learned solution $A$ to the relaxation. By sampling each layer's bit-width from the learned distribution $A^l$, randomized rounding captures uncertainty in the super-network's preferences. Another simple deterministic strategy is to select, for each layer, the quantization option with maximum probability [4], where the data format with the largest associated parameter is sampled, finally others also propose the use of the Gumbel-Softmax [28, 29] to simulate random categorical sampling steps during the search phase and enforce the convergence of the distribution by scheduling the "temperature" hyperparameter of the Gumbel-Softmax function.

# 3 Constrained Bit Allocation

***Post-training compression.*** While neural architecture search for mixed-precision models has mostly been developed as part of a quantization-aware training framework, recent investigations [15] have shown that the optimal approach to model compression is to first train large networks in full precision, and then aggressively compress the model for deployment. By following this framework, we rework on the assumptions of the neural network bit-allocation problem. We propose to restrict the problem to the search of architectural parameters for pre-trained models, maintaining the previously learned weights $W^*$ frozen. This significantly reduces the number of parameter updates, as the number of architecture parameters only grows linearly with the depth of the associated network and is invariant with respect to the dimensions of its layers.

$$\begin{aligned}
\underset{A}{minimize} \quad & \mathcal{L}(A, \hat{W}^*) \\
subject\ to \quad & \sum_{d=1}^{D} A_d^l = 1 \quad \forall l \in \mathbb{L} \\
& A_d^l \geq 0 \quad \forall (l, d) \in \mathbb{L} \times \mathbb{D}
\end{aligned} \quad (5)$$

***User-defined architectural constraints.*** In practical deployment scenarios, models must conform to diverse hardware constraints, including limits on total model size, supported numerical formats, and compute budgets such as FLOPs or BOPs. These constraints are platform-dependent and are often non-negotiable. To accommodate such deployment requirements, we reformulate the bit allocation task not as a multi-objective trade-off between accuracy and complexity, but as a constrained optimization problem. Let $C(A, W)$ be a differentiable architectural cost function (e.g., total model size in bits), and let $B$ denote an upper bound imposed by the hardware. Our goal is to minimize the loss subject to this constraint:

$$\begin{aligned}
\underset{A}{minimize} \quad & \mathcal{L}(A, \hat{W}^*) \\
subject\ to \quad & \mathcal{R}_B[A] = B - C(A, W) \geq 0 \\
& \sum_{d=1}^{D} A_d^l = 1 \quad \forall l \in \mathbb{L} \\
& A_d^l \geq 0 \quad \forall (l, d) \in \mathbb{L} \times \mathbb{D}
\end{aligned} \quad (6)$$

This constrained formulation enables the principled integration of hardware-awareness into the precision allocation process, ensuring that the resulting architecture is both accurate and deployable.

***Interior-Point Formulation.*** To solve the non-linear constrained optimization problem defined in Equation equation 6, we employ techniques common in constrained optimization, specifically adopting a regularizer-based interior-point method. The first step involves formulating the Lagrangian function, which incorporates the objective function and the constraint scaled by a Lagrange multiplier $\lambda$:

$$\mathcal{L}_\lambda(A) = \mathcal{L}(A, W^*) + \lambda \mathcal{R}_B[A] \quad (7)$$

Here, $\mathcal{L}(A, W^*)$ represents the original loss function (our objective to minimize) with fixed model weights $W^*$, and $\mathcal{R}_B[A]$

represents the hardware constraint function (which must be non-negative, $\mathcal{R}_B[A] \geq 0$). The variable $\lambda$ is the Lagrange multiplier associated with this inequality constraint.

For a given candidate solution $A^*$ to be a local optimum of the constrained problem equation 6 (under certain regularity conditions), it must satisfy the Karush-Kuhn-Tucker (KKT) conditions. These conditions are fundamental necessary conditions for optimality in nonlinear programming [19]. They generalize the method of Lagrange multipliers to handle inequality constraints. For our problem, the KKT conditions are:

$$\mathbf{KKT} \begin{cases}
\nabla_A \mathcal{L}(A^*, W^*) + \lambda \nabla_A \mathcal{R}_B[A^*] = 0 \quad \text{(Stationarity)} \\
\mathcal{R}_B[A^*] \geq 0 \quad \text{(Primal Feasibility)} \\
\lambda \geq 0 \quad \text{(Dual Feasibility)} \\
\lambda \mathcal{R}_B[A^*] = 0 \quad \text{(Complementary Slackness)}
\end{cases} \quad (8)$$

However, in practice, satisfying the strict complementarity condition $\lambda \mathcal{R}_B[A^*] = 0$ leads to optimization challenges due to discontinuity at the boundary of the feasible region. To circumvent this, we adopt the perturbed KKT formulation, commonly used in interior point methods, which replaces the complementarity condition with a small nonzero slack.

$$\mathbf{KKT}(\mu) \begin{cases}
\nabla \mathcal{L}(A^*, \hat{W}^*) + \lambda \nabla \mathcal{R}_B[A^*] = 0 \\
\mathcal{R}_B[A^*] \geq 0 \\
\lambda \geq 0 \\
\lambda \mathcal{R}_B[A^*] = -\mu
\end{cases} \quad s.t. \quad \mu \to 0 \quad (9)$$

This relaxation smooths the boundary behavior of the optimizer and permits convergence to the constrained optimum from within the feasible set. It corresponds to minimizing the following regularizer-augmented objective:

$$\underset{A}{minimize} \quad \mathcal{L}(A, \hat{W}^*) - \mu \log(\mathcal{R}_B[A]) \quad (10)$$

Here, the logarithmic regularizer $\log(\mathcal{R}_B[A])$ diverges to $-\infty$ as the constraint approaches 0, effectively discouraging the optimizer from leaving the feasible region.

***Algorithm.*** Following our previous derivation, we propose the following iterative algorithm.

---

**Algorithm 1** Constrained Bit Allocation

---

**Require:** Pretrained weights $W^*$, initial allocation $A$, loss function $\mathcal{L}(\cdot, \cdot)$, constraint $\mathcal{R}_B[\cdot]$, initial regularizer weight $\mu > 0$, decay factor $0 < \delta < 1$, number of iterations $T$

**Ensure:** Final discrete allocation $\hat{A}$ satisfying $\mathcal{R}_B[\hat{A}] \leq B$

1: $t \leftarrow 1$
2: $\mu^{(1)} \leftarrow \mu$
3: **while** $t \leq T$ **do**
4: $\quad A^{(t)} \leftarrow \arg\min_A \mathcal{L}(A, W^*) - \mu^{(t)} \ln(\hat{\mathcal{R}}_B[A])$
5: $\quad \mu^{(t+1)} \leftarrow \delta \mu^{(t)}$
6: $\quad t \leftarrow t + 1$
7: **end while**
8: $\hat{A} \leftarrow \text{round}(A^{(T)})$
9: **return** $\hat{A}$

---

The optimizer we use in this paper is ten epochs of the classic back-propagation algorithm already provided by all the most popular Machine Learning libraries, however other optimization algorithms could also be considered. In particular, Newton's method is often the optimizer of choice for interior-point methods such as the one discussed here. The rounding step depends on the sampling approach. In this work, we follow the multi-nomial sampling approach as described earlier.

To minimize the model loss while staying within region defined by the memory constraint, the interior-point method adds a logarithmic regularizer penalty. This penalty acts like a repulsive force that becomes very large at the constraint boundary, ensuring the solution always remains strictly feasible. Optimization starts with a strong repulsion (large $\mu$), keeping the solution within the feasible region. As $\mu$ is gradually decreased across iterations, the regularizer's influence weakens, allowing the solution to follow a "central path" closer to the true minimum of L while still being repelled from the boundary. The figure below illustrates how each step balances minimizing the objective (descent step) with staying feasible (step towards central path), ultimately converging to the constrained optimum as $\mu \rightarrow 0$.

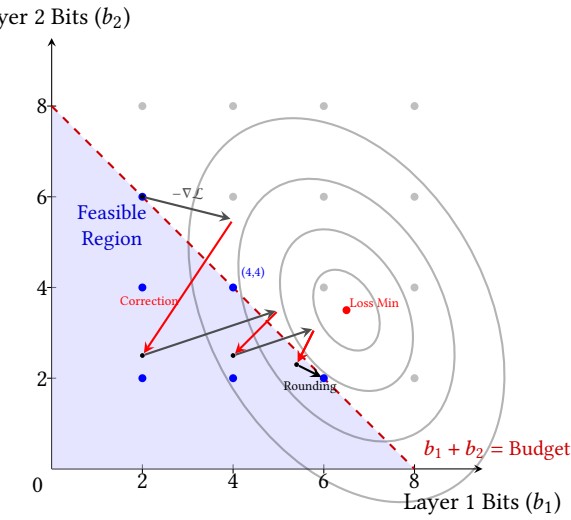

**Figure 1: Search space for bit allocation in a 2-layer network. Axes are bits per layer ($b_1, b_2$). Blue dots are feasible discrete choices under a budget constraint (shaded region, $b_1 + b_2 \leq$ Budget). Gray dots are infeasible. Gray ellipses represent level curves of the loss function, with the true (continuous) minimum marked in red, located between discrete points.**

## 4  Experimental Methodology and Results

To evaluate the effectiveness of our method, we conducted a series of post-training experiments on autoregressive transformer models of varying scales, including LLaMA [11], Gemma [24, 25], and Qwen [22, 26, 27] architectures (specifically Llama-3.2, Gemma, and Qwen2.5 models) ranging up to 3 billion parameters. All experiments follow the setup for causal language modeling provided by

the open-source examples of HuggingFace. Importantly, we operate strictly in the post-training regime: the model weights are frozen, and only the architectural parameters governing format assignment are optimized using a small calibration set. All models are adapted using a mixture of MX-compliant formats, specifically MXFP4 and MXFP8.

***Empirical study 1: few-shot scenario.*** In the few-shot setting, we use a small calibration set of 128 samples drawn from the training split of the C4 corpus [8]. We then evaluate perplexity (PPL) on the C4 validation split, gauging performance on in-distribution data under low-data conditions. Figure 2 compares the perplexity of our mixed-precision allocations constrained to an average of 4.5 bits ('Mixed', red 'x') against uniform MXFP baselines.

We observe significant improvements for several models with only a minor increase in average bits compared to uniform MXFP4. Specifically, the mixed allocation yields perplexity drops of over 3 points for Gemma-3-1B-it ($\downarrow$3.67), 9 points for Qwen2.5-1.5B ($\downarrow$9.06), and 7 points for Qwen2.5-3B ($\downarrow$7.46), as indicated by the black arrows in Figure 2. These improvements are achieved with only approximately +0.25 effective bits compared to the MXFP4 baseline (effective bits $\approx$ 4.25). Notably, for these models, the mixed 4.5-bit allocation outperforms even the uniform MXFP6 and MXFP8 baselines, despite using substantially fewer bits. For the remaining models (Qwen2.5-0.5B, Llama-3.2-1B, Gemma-2B), the mixed 4.5-bit version achieves performance closely matching or slightly better than the uniform MXFP6 baseline, again offering considerable bit savings.

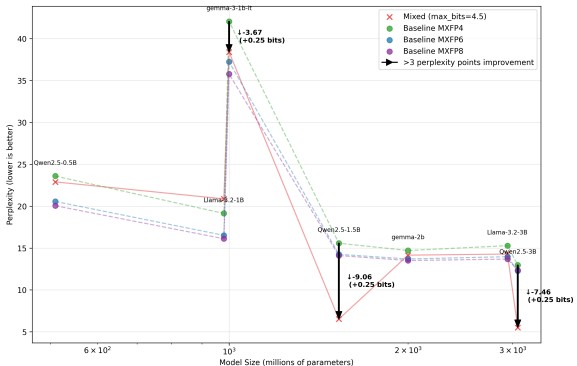

**Figure 2: Few-shot perplexity (lower is better) on C4 validation set for models constrained to max_bits=4.5. Our mixed allocation (red 'x') is compared against uniform MXFP4, MXFP6, and MXFP8 baselines. Black arrows indicate perplexity improvements >3 points over the MXFP4 baseline.**

Figure 3 shows the achieved effective bit-widths for the models under the 4.5-bit constraint in this few-shot setting. The results demonstrate that the target average bit-width constraint is closely adhered to across all models. Most resulting effective bit-widths match the constraint or show very minor deviations, typically less than 0.1 bits higher than the enforced upper bound, confirming the framework's ability to precisely manage the hardware budget.

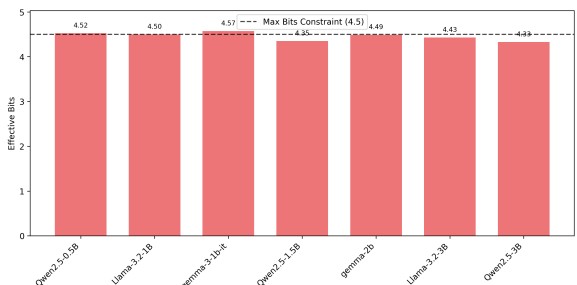

**Figure 3: Effective bits achieved for models constrained to max_bits=4.5 in the few-shot setting. The dashed line indicates the target constraint.**

***Empirical study 2: zero-shot scenario.*** For zero-shot evaluation, we calibrate using 256 samples from the C4 training split. We then employ the 'lm-eval-harness' benchmark suite [1, 10] to assess model performance on downstream tasks without task-specific fine-tuning. We report accuracy on LAMBADA [21] (predicting the last word of a passage, testing context understanding) and MMLU [12] (multitask accuracy across diverse subjects).

Figure 4 presents LAMBADA accuracy results for the 4.5-bit constraint. Compared to the uniform MXFP4 baseline, our mixed allocation provides significant accuracy boosts with only $\approx$ +0.25 more effective bits: +9.8% for Qwen2.5-0.5B, +2.4% for Llama-3.2-1B, +3.8% for Gemma-3-1B-it, and +6.0% for Gemma-2B. For the Gemma models in particular, the mixed 4.5-bit versions achieve LAMBADA accuracy very close to that of the uniform MXFP6 baseline (effective bits $\approx$ 6.25), demonstrating substantial efficiency gains.

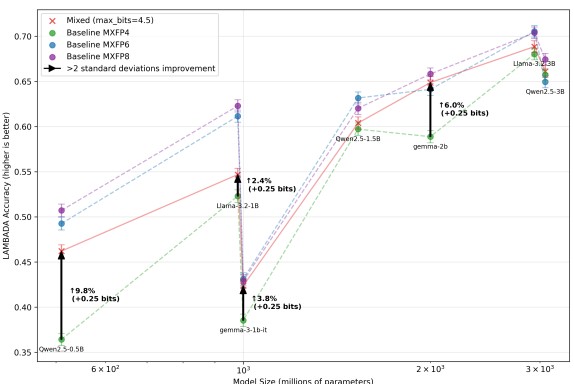

**Figure 4: Zero-shot LAMBADA accuracy (higher is better) for models constrained to max_bits=4.5. Our mixed allocation (red 'x') compared against uniform baselines. Black arrows indicate accuracy improvement over MXFP4 baseline exceeding 2 standard deviations.**

We further evaluate on MMLU (results shown in Figure 5). Here too, the mixed 4.5-bit allocation consistently improves accuracy over the MXFP4 baseline, with gains ranging from +1.2% up to +4.0% across the different models. It is crucial to note that these accuracy improvements on both LAMBADA and MMLU are achieved even

though the model weights remain frozen and the architectural parameters were calibrated solely on C4 data, highlighting the generalization capability of the learned allocations.

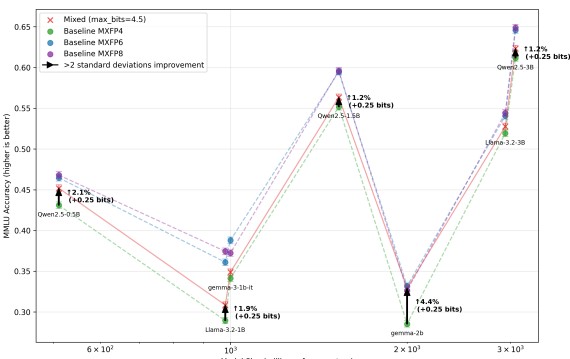

**Figure 5: Zero-shot MMLU accuracy (higher is better) for models constrained to max_bits=4.5. Our mixed allocation (red 'x') compared against uniform baselines.**

Figure 6 again confirms that the effective bit-widths achieved in the zero-shot scenario under the 4.5-bit constraint adhere closely to the target, reinforcing the method's precise budget control.

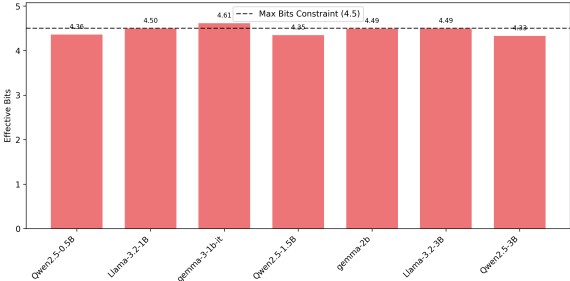

**Figure 6: Effective bits achieved for models constrained to max_bits=4.5 in the zero-shot setting. The dashed line indicates the target constraint.**

These zero-shot findings underscore that targeted precision allocation, guided by our constrained optimization, effectively preserves crucial model capabilities for downstream tasks, often significantly better than uniform low-bit formats, particularly under tighter memory constraints. The method provides practitioners fine-grained control over the accuracy-compression trade-off simply by selecting the appropriate average bit-width constraint.

## 5 Discussion

The experimental results presented herein demonstrate the effectiveness of our constrained optimization framework for post-training mixed-precision allocation. By leveraging an interior-point method with a logarithmic barrier, our approach successfully navigates the complex trade-off between model performance (perplexity and accuracy) and computational constraints (represented by average bit-width, directly correlating with memory footprint). Across

 

various model architectures and scales, the learned mixed-precision configurations consistently outperform uniform low-bit baselines, often achieving performance comparable to high-precision formats while operating at significantly lower effective bit-widths.

A key strength of this methodology lies in its practicality and efficiency. The entire allocation process operates post-training, requiring no modification or fine-tuning of the original model weights. This significantly reduces the computational cost and technical complexity typically associated with model adaptation. Furthermore, the optimization relies on a remarkably small number of calibration samples (e.g., 128-256), making it feasible even when access to large datasets is limited. Despite this minimal overhead (e.g., approximately 5 minutes for a 500M parameter model on one NVIDIA A100 GPU), the derived allocations exhibit robust generalization, improving performance not only on in-distribution validation data but also on diverse, unseen zero-shot tasks such as OpenAI's LAMBADA. This suggests that the allocation, guided by gradients on the calibration set, effectively identifies and preserves critical computational pathways within the network.

From a theoretical standpoint, formulating the problem as a constrained optimization provides a more rigorous foundation compared to heuristic methods or multi-objective optimization techniques that may lack strong guarantees on the output mixed precision scheme or may require complex Pareto front analysis. The interior-point method is well-established, and its application here, using the average bit-width constraint, offers a natural and interpretable way to incorporate hardware limitations, particularly memory capacity, directly into the optimization objective. This explicit constraint mechanism grants practitioners fine-grained control over the desired operating point on the accuracy-efficiency curve. The hyperparameters associated with the interior-point method are relatively few and possess clear interpretations within the optimization context, facilitating tuning.

We acknowledge certain limitations. The current approach implicitly utilizes a "supernet" concept during the search phase, where gradients for different format assignments are needed. This can temporarily increase memory usage during the allocation optimization compared to standard inference. However, several factors mitigate this concern. Firstly, the search often involves low-bit formats (e.g., MXFP4, MXFP8); the memory required to hold activations or gradients for multiple low-bit options might still be comparable to, or less than, holding a single higher-precision (e.g., FP16 or BF16) baseline tensor. For instance, exploring two sub-8-bit formats could theoretically fit within the space of one FP16 tensor. Secondly, the number of parameters being optimized during allocation scales only with the number of assignable layers or modules (model depth), which is orders of magnitude lower than the total number of weights in the LLM. This makes the gradient updates for the allocation parameters substantially less demanding than full model training.

Nonetheless, optimizing the memory of the supernet and the computational efficiency of the allocation search itself presents a viable avenue for future research. Exploring techniques like shared projection heads during the search, path Gumbel-Softmax, or more advanced gradient estimation methods could potentially reduce the overhead further. Investigating the interplay between different types of constraints (e.g., latency-aware constraints) within this framework is another promising direction.

## 6 Conclusion

Efficient deployment of complex DNNs on resource-constrained hardware is crucial. While mixed-precision formats offer potential, determining the optimal layer-wise allocation under hardware constraints remains challenging, often addressed by heuristics or complex training-aware searches. This paper introduces a principled post-training framework for mixed-precision allocation grounded in constrained optimization. Using an interior-point method, our approach explicitly incorporates hardware limitations, like average bit-width constraints, directly into the allocation process. This operates efficiently post-training, requiring only small calibration datasets and no costly retraining. Empirical evaluations across diverse transformer architectures (including Llama-3.2, Gemma, Qwen2.5) demonstrate effectiveness. Learned allocations consistently outperform uniform low-bit baselines in few-shot and zero-shot scenarios, showing robust generalization. The method offers practitioners fine-grained control over the performance-efficiency trade-off. We demonstrated that strategic allocation, even using only the lowest and highest available precisions (e.g., MXFP4/8), yields compelling compression solutions. In summary, our constrained optimization framework provides a theoretically sound, data-efficient, and computationally inexpensive method for hardware-aware mixed-precision allocation. It is a valuable tool for deploying large models effectively under tangible resource constraints, bridging the gap between model capability and practical feasibility.

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
