# OpenReview forum: "Constrained bit allocation for mixed-precision deep neural networks"
_iscaconf.org/ISCA/2025/Workshop/MLArchSys — MLArchSys 2025 Oral_

### Official Review · Reviewer_LDz2 · 2025-05-17
**interesting method but lacking relevant comparisons**

**Confidence:** 4
**Rating:** 5

**Detailed Feedback And Questions For Authors:**

Quality: The authors develop their method in a principled way and apply it across a good range of different settings. However, the choice of average bit width as the optimization target and some results indicating the method cannot guarantee adherence to constraints negatively impact the overall quality.

Clarity: The paper is clearly written and easy to follow. Figures are well-designed and effectively help the reader understand the proposed method.

Originality: While previous papers have addressed this problem, applying the interior point method in this specific context is, to the best of my knowledge, novel.

Significance: Given the missing comparisons to other post-training mixed-precision approaches and the unstated computational cost, it's challenging to assess whether the method delivers meaningful improvements in practice.

**Top Reasons To Accept The Paper:**

- Problem Relevance and Setup: The paper effectively addresses a relevant problem: post-training quantization under hardware constraints. The authors clearly outline challenges with current approaches, such as the lack of guarantees that hardware constraints will be met, the significant data requirements for some methods to work well, the risk of modifying models weights in quantization aware training approaches, and the difficulty in granularly controlling bit precision (combinatorial complexity).
- Methodology Clarity: The methodology section is well-structured, clearly explaining and illustrating (easy to understand figure and algorithm) the optimization problem in a sensible manner.
- Quality Improvements: The proposed method demonstrates noticeable quality improvements over uniform quantization with only minimal overheads over selected baselines.
- Rigorous Evaluation: The authors conducted a rigorous evaluation of their method across various model architectures, sizes, and tasks, which strengthens the paper.
- Novelty of optimization method: the interior point method in this context appears to be novel.

**Top Reasons To Reject The Paper:**

- Missing Baselines: The paper lacks a comparison against other post-training mixed-precision quantization approaches. Some of these methods can perform bit allocations without any data [1,2,3,4].
- Hardware Constraint Adherence: While the authors highlight that other methods may not guarantee hardware compatibility, their own method sometimes appears to exceed the stated maximum bit constraint (e.g., Figure 3 shows Gemma-3-1B-IT at 4.57 bits, and similar instances are visible in Figure 6).
- Optimization Target: The optimization target of minimizing average bit width seems only loosely connected to practical hardware constraints. A more relevant target might involve imposing limits on the total model size in bits *and* the maximum single-layer size in bits, which would better reflect the capacity of memory hierarchies of target platforms.
- Computational Cost: The paper does not discuss the computational expense of the optimization method. It would be valuable to understand how long this process takes to meaningfully compare its advantages against potentially higher-quality but more computationally intensive quantization-aware training methods.

References:
[1] https://arxiv.org/pdf/2001.00281
[2] https://arxiv.org/pdf/2306.04879
[3] https://proceedings.mlr.press/v139/hubara21a.html
[4] https://arxiv.org/abs/2011.10680

---

### Official Review · Reviewer_GLLK · 2025-05-18
**Constrained bit allocation for neural networks Review**

**Confidence:** 4
**Rating:** 6

**Detailed Feedback And Questions For Authors:**

- Formatting: opening quotations in Latex can use \` instead of ', e.g. ’x’ -> \`x\'
- In the abstract (improving performance by up to several standard deviations in some cases): The number of standard deviations suggest a statistically significant result, however the values measured will be dependent on the amount of data used for the evaluation. I would suggest pointing out some of the other metrics instead, such as the perplexity change.
- Figure 1: I understand this is meant as an illustration, but in the way the problem is formulated, would we expect to actually have a loss minimum for some tuple (b1, b2) of bit values? Wouldn't the loss continue to improve with arbitrarily large bit values?
- Why was the average bitwidth of 4.5 selected? What do the results look like for other values (e.g. 5?). It would be especially interesting to use 6 because it can be compared directly with the uniform MXFP6 format.
- I am curious about the probabilities (the values before rounding) for the selected formats. Do they tend to be close to the extremes, or are there many values in between? How much variability does the sampling introduce, and how much does this matter?

Suggestions for future work:
- Instead of constraining to MX block formats, consider more general ExMy floating point formats, additionally with different block sizes (and perhaps other quantization parameters). There is no fundamental reason to use blocks of size 32, especially since this is focused on compression, and you may see better compression with other formats.

**Top Reasons To Accept The Paper:**

The paper formulates the post-training compression problem of weight tensors given a user-defined average bit-width constraint, and approaches finding a solution using an interior-point formulation of the optimization problem. The optimization involves perturbing the KKT conditions using a regularization weight parameter to improve the convergence properties. The authors demonstrate using a target value of 4.5 bits and closely approaching that value using a mixture of MXFP4 and MXFP8 formats, and show experimental results comparing the mixed 4.5 format for 4, 6, and 8. This is an interesting and novel approach to model compression.

**Top Reasons To Reject The Paper:**

The main thing that looks a little weird to me is that the mixed strategy (MXFP4 / MXFP8) can sometimes beat the MXFP8. This doesn't make sense to me, because the MXFP8 should be strictly better. This could be due to noise, or other factors. I hope the authors are able to address this, e.g. maybe there is an explanation of why it can happen that I don't understand, or maybe more data needs to be collected.

Another sanity check could be negating the loss function except for the constraint, and checking if the results are significantly different.

---

### Official Review · Reviewer_Xp7H · 2025-05-18

**Confidence:** 5
**Rating:** 5

**Detailed Feedback And Questions For Authors:**

1. Must cite recent papers [1,2] in related work section of post training compression.

[1] Ramachandran, A., Wan, Z., Jeong, G., Gustafson, J., & Krishna, T. (2024, June). Algorithm-hardware co-design of distribution-aware logarithmic-posit encodings for efficient dnn inference. In Proceedings of the 61st ACM/IEEE Design Automation Conference (pp. 1-6).

[2] Ramachandran, A., Kundu, S., & Krishna, T. (2024). MicroScopiQ: Accelerating Foundational Models through Outlier-Aware Microscaling Quantization. arXiv preprint arXiv:2411.05282.

**Top Reasons To Accept The Paper:**

The paper introduces a rigorous constrained optimization formulation for post-training mixed-precision bit allocation, departing from previous heuristic or soft-penalty-based approaches and requires no retraining. The framework explicitly integrates hardware constraints (e.g., bit budgets) into the optimization problem using an interior-point method.

**Top Reasons To Reject The Paper:**

The current experiments focus primarily on perplexity and accuracy under bit constraints. While memory is addressed via effective bits, there’s no detailed analysis of real hardware latency, throughput, or energy implications.

While MXFP formats are used effectively, the study only explores two discrete format options (MXFP4 and MXFP8). It remains unclear how well the framework generalizes across broader format choices.

---

### Official Review · Reviewer_htNQ · 2025-05-21
**Mixed-precision allocation with constrained optimization**

**Confidence:** 3
**Rating:** 6

**Detailed Feedback And Questions For Authors:**

The paper presents a differentiable constrained optimization approach for mixed-precision quantization that adapts DNAS principles with average bit size constraints. The methodology is clear and results are promising.

Questions
- What is the accuracy achieved if the max bit size is set to 4 instead of 4.5. Will it outperform mxfp4?
- Could you replace proxy bit size with actual hardware performance metrics (latency/throughput) as direct optimization constraints?
- Could this framework be extended to optimize communication overhead in low-precision training scenarios?

**Top Reasons To Accept The Paper:**

+ New DNAS formulation with NN bit allocation set as constraint
+ Efficient post-training technique without the need for retraining
+ Quantized models closely follow the target average bit constraints
+ Constant accuracy improvement over mxfp4

**Top Reasons To Reject The Paper:**

- Lack of empirical evaluation for real hardware performance